# Anticipatory Antifungal Treatment in Critically Ill Patients with SARS-CoV-2 Pneumonia

**DOI:** 10.3390/jof9030288

**Published:** 2023-02-22

**Authors:** Ángel Estella, María Recuerda Núñez, Carolina Lagares, Manuel Gracia Romero, Eva Torres, Juan Carlos Alados Arboledas, Álvaro Antón Escors, Clara González García, Dolores Sandar Núñez, Dolores López Prieto, Juan Manuel Sánchez Calvo

**Affiliations:** 1Intensive Care Unit, Medicine Department, University of Cádiz, University Hospital of Jerez, INIBiCA, 11407 Jerez, Spain; 2Department of Statistical and Operational Research, PAIDI Group CTS553, University of Cádiz, 11510 Cádiz, Spain; 3Microbiology Department, University Hospital of Jerez, 11407 Jerez, Spain; 4Medicine Faculty of Cádiz, University of Cádiz, 11510 Cádiz, Spain

**Keywords:** SARS-CoV-2, liposomal amphotericin B, ARDS, aspergillus, bronchoalveolar lavage, ICU

## Abstract

Background. The aim of this study was to investigate the incidence of COVID-19-associated pulmonary aspergillosis (CAPA) in critically ill patients and the impact of anticipatory antifungal treatment on the incidence of CAPA in critically ill patients. Methods. Before/after observational study in a mixed intensive care unit (ICU) of a university teaching hospital. The study took place between March 2020 and June 2022. Inclusion criteria were critically ill patients with severe SARS-CoV-2 pneumonia requiring invasive mechanical ventilation. Two analysis periods were compared according to whether or not antifungal therapy was given early. Results. A total of 160 patients with severe SARS-CoV-2 pneumonia and invasive mechanical ventilation were included. The incidence of CAPA in the first study period was 19 out of 58 patients (32.75%); during the second period, after implementation of the intervention (anticipatory antifungal therapy), the incidence of CAPA decreased to 10.78% (11 out of 102 patients). In patients with CAPA under invasive mechanical ventilation, the mortality rate decreased from 100% to 64%. Conclusions. Anticipating antifungal treatment in patients with SARS-CoV-2 pneumonia under invasive mechanical ventilation was associated with a decrease in the incidence and mortality of pulmonary aspergillosis.

## 1. Introduction

Invasive pulmonary aspergillosis (IPA) is a serious infectious complication in critically ill patients, and it is associated with very high mortality rates, especially among mechanically ventilated patients [1]. Published guidelines highlight the importance of early action [2]; however, to avoid overtreatment, it is essential to identify patient risk factors to improve diagnostic accuracy. IPA has been classified as an opportunistic infection that mainly affects immunocompromised patients with neutropenia or undergoing hematopoietic transplantation [3]. In recent years, some reports have also shown an increased incidence in critically ill patients with chronic respiratory lung disease, especially those on corticosteroid therapy [4]. The 2009 influenza A H1N1v pandemic led to an increased incidence of viral pneumonia in ICUs, and multicenter studies reported that there was a marked increase in the incidence of IPA in patients with acute respiratory distress syndrome (ARDS) [5,6].

COVID-associated pulmonary aspergillosis (CAPA) has been reported with great variability, with low incidence reports [7] compared to other data estimating that IPA occurs in up to 30% of cases [8]. These differences are probably due to the different diagnostic protocols applied [9]. In the last two years, the emergence of COVID-19 has been a major shock to healthcare systems worldwide. The pandemic has highlighted that 10–20% of hospital admissions must be admitted to the ICU due to the development of ARDS [10]. Mortality in these patients is unacceptably high [11] and the development of secondary infection, including IPA, has been the subject of debate. We hypothesized that a strategy of early administration of antifungal treatment would decrease the development of CAPA and ultimately improve patient survival. The aim of this study was to describe the incidence of CAPA in critically ill patients and the effect of early antifungal therapy on the incidence of CAPA.

## 2. Materials and Methods

The study was conducted in a mixed ICU in a university teaching hospital. The study period was from March 2020 to June 2022. Inclusion criteria were patients with severe SARS-CoV-2 pneumonia requiring invasive mechanical ventilation and bronchoscopic evaluation. These patients were considered an at-risk population for CAPA. Exclusion criteria were patients whose clinical and/or hemodynamic status did not guarantee that fibrobronchoscopy could be performed safely. More specifically, the exclusion criteria were: (1) endotracheal tube size less than 7.5 mm (internal diameter); (2) pneumothorax identified on chest X-ray prior to fibrobronchoscopy; (3) oxygen saturation by pulse oximetry less than 80%, with FIO2 of 1; (4) severe acidosis (pH < 7.20); (5) hemodynamic instability, defined as systolic blood pressure <90 mmHg despite vasoactive drugs. On admission to the ICU, diagnostic fibrobronchoscopy was performed in all mechanically ventilated patients. Bronchoscopic bronchoalveolar lavage (BAL) was performed with 150 mL of physiological saline solution, divided into three aliquots. The first 20 mL of BAL was discarded, and a sample of the remaining fluid was collected for microbiological analysis. Fibrobronchoscopy was performed with disposable AMBU^®^ aScope™ 4 Broncho Regular OD 5.8/2.8 mm bronchoscopes (Copenhagen, Denmark). The bronchoscope was inserted using a special adapter valve to prevent aerosol generation during the procedure, which was performed in pressure control ventilation mode, maintaining the optimal PEEP level determined previously, and a fraction of inspired oxygen of 100%. After the procedure, FIO2 was adjusted to the patient’s clinical status. The patient was not repositioned to supine position during the diagnostic procedure. The continuous infusion sedative being used to sedate the patient was not changed.

During bronchoalveolar lavage, the bronchoscope was directed either to the location suggested on the chest radiograph, to the area most affected according to direct visualization, to the middle right lobe in case of bilateral infiltrates if the patient was in supine position, or to the lower left lobe if the patient was in prone position. No local anesthetics were administered, and aspiration through the bronchoscope channel was not attempted until it was in position at the site to be studied. In the laboratory, samples were stained directly (by Gram stain) to detect mycobacteria, as well as fungi (using calcofluor white stain) and the BAL sample was processed for a quantitative microbiological culture. The presence of *Aspergillus* galactomannan antigen was determined. RT-PCR for detection of the Influenza A (H1N1) virus was performed using the MagNA Pure Compact kit (Real Time ready Influenza A H1N1 Detection Set, Roche Applied Science, Mannheim, Germany), and also for detection of SARS-CoV-2.

Due to the lack of specificity for the diagnosis of pulmonary aspergillosis in critically ill patients and the absence to date of established clinical criteria for patients with SARS-CoV-2, diagnosis was based on positive molecular testing (PCR) and the presence of relevant clinical criteria, defined as fever and/or impaired respiratory function, presence of pulmonary infiltrates on a simple chest radiograph, and mycological criteria for infection based on a good quality invasive respiratory specimen obtained by fibrobronchoscopy. Calcofluor white staining (Becton Dickinson, Franklin Lakes, NJ, USA) was performed after bronchoscopy. Bronchoalveolar lavage (BAL) specimens were cultured in sabouraud agar with chloramphenicol (Becton Dickinson). Galactomannan was measured using the Platelia *Aspergillus* galactomannan assay (Bio-Rad Laboratories, Hercules, CA, USA) with a GM cut-off value of <0.5 as a useful negative predictive value to rule out aspergillosis [12], thus determining the presence of fungus by calcofluor white staining and/or fungal culture. The variables analyzed were clinical characteristics, laboratory test, ICU length of stay, mortality, previous administration of corticosteroids. We distinguished two periods of information collection: a first period, from March 2020 to November 2020, when no antifungal treatment was administered on admission to the ICU, and a second period, from November 2020 to June 2022, when the clinical care team, concerned about the documented high incidence of aspergillosis, decided to treat patients with severe SARS-CoV-2 pneumonia early with antifungals until the presence of aspergillus was ruled out, in which case antifungal treatment was suspended (Figure 1).

Amphotericin B was the preferred choice based on its demonstrated efficacy, safety profile, broad spectrum for fungal disease, and also to reserve the first-line treatment for cases of confirmed fungal infection. Liposomal amphotericin B was administered at a dose of 3 mg/kg/day for the first three days, pending the results of fungal microbiological testing. Because of the nature of this pandemic disease and the contact restrictions in place, informed consent for the bronchoscopy and treatment approach was obtained from the patient whenever possible, from family members if they were in the hospital, or by phone call, and was recorded in the medical history. Patient data were recorded in the SAMIUCCOVID registry, approved by the research ethics committee.

Statistical analysis. For qualitative variables, a descriptive analysis was carried out, using absolute frequencies and percentages. For quantitative variables, we checked whether they followed normal distributions, and were summarized using median and 25th and 75th percentiles. For comparisons between treatment subgroups and/or the presence of CAPA, the chi-square test for qualitative variables was performed, and the non-parametric Mann–Whitney test for quantitative variables. Finally, multivariate survival analysis using the Cox regression model was performed.

## 3. Results

During the study period, bronchoalveolar lavage was performed on admission in 160 mechanically ventilated patients with severe SARS-CoV-2 pneumonia. The microbiology laboratory was asked to actively search for CAPA, and the incidence of proven CAPA was 18.75%. The mean age of patients was 65 years old, and 72% were male. In the control study period, the incidence of CAPA was 32.75%, decreasing to 10.78% in the intervention study period when all patients received early antifungal treatment at admission pending microbiological confirmation. The clinical characteristics of the patients are shown in Table 1.

There were no differences in the main laboratory parameters between groups, except for the inflammation parameters, which were higher in patients who did not receive early antifungal treatment, but who, nevertheless, had higher D-dimer levels (Table 2).

With respect to concomitant treatments received, significant differences were observed between the two periods analyzed in the use of corticosteroids: 78% in patients not treated with antifungals versus 94% in those pretreated with antifungal therapy; 3% of patients without antifungal therapy received treatment with tocilizumab versus 27% in the group given early antifungal therapy. No treatment-related adverse effects on renal function were observed. Table 3 shows the significant differences in pulmonary aspergillosis and mortality between the study periods due to early antifungal therapy in patients with severe SARS-CoV-2 pneumonia.

APACHE II and SOFA scores at admission and at the beginning of mechanical ventilation with laboratory tests were also analyzed in each group according to the occurrence of CAPA (Table 4); there were no significant differences except in mortality and severity. Values are shown in frequencies and percentages or quartiles, depending on whether the variable was numerical or not.

Finally, Cox regression was performed to assess patient survival according to study group and the appearance of CAPA. The final model included, as covariates that directly affect the survival of these patients, the appearance of pulmonary aspergillosis together with age. The model discarded other clinical variables previously analyzed. The occurrence of CAPA had an odds ratio (OR) of 1.732 (95% CI 1.081–2.773) (*p* = 0.022), and age an OR of 1.055 (95% CI 1.03–1.08), with statistical significance of 0.000. In Figure 2, the overall survival curve shows longer survival in patients who received early antifungal therapy. Further study of these curves also reveals improved survival in patients who do not develop CAPA (Figure 3a,b).

## 4. Discussion

Diagnosis of fungal infection in an ICU remains a challenge. Since the symptoms and radiological pattern in immunocompetent patients are nonspecific, a high index of clinical suspicion is required to consider testing for aspergillosis. If this possibility is not taken into account, the disease is likely underdiagnosed [9]. The main novel finding of this study is the high incidence of pulmonary aspergillosis detected using an invasive diagnostic strategy, and the decreased incidence following early antifungal treatment based on a high suspicion of aspergillosis with the presence of risk factors for this infection: SARS-CoV-2 pneumonia, ARDS, corticosteroid and/or immunomodulatory therapy. Although clinical experiences of pulmonary aspergillosis in patients with SARS-CoV-2 have been documented in the literature [13,14,15,16,17,18], the microbiological diagnosis in this study was based on an invasive respiratory sample obtained by fibrobronchoscopy, on the grounds that, if we were trying to distinguish between colonization and infection, a targeted sample from the alveolar space would be of higher quality than one taken from the upper respiratory tract. Interestingly, aspergillosis was diagnosed on admission to the ICU, not during prolonged admission; this supports the theory that it is the viral infection in its early stages that produces a state of immunoparalysis or lymphocyte dysfunction that favors aspergillus infection. In the early stages of the SARS-CoV-2 pandemic, the lack of diagnoses with invasive samples in this patient population can be explained by the initial pandemic recommendation to avoid invasive procedures such as bronchoscopy due to the risk to the operator of generating aerosols [19,20], particularly in the early days of admission to the ICU when the viral load is assumed to be greater [21,22]. In the course of the study period, the high incidence of aspergillosis observed in the early months of the pandemic led us to report the documented incidence to the Preventive Medicine Department of our hospital and to study the presence of spores in different environmental samples. During the pandemic, most hospitals around the world undertook work to increase available space and made structural alterations to adapt existing areas to increase health care capacity, which, as is well known, increases the concentration of fungal spores in the environment. This was ruled out by tests carried out by the Preventive Medicine Department, which failed to demonstrate the presence of spores in the environment that would justify our high incidence of aspergillosis. Since patients with SARS-CoV-2 pneumonia are not a priori considered to be immunocompromised, the classic diagnostic criteria of the EORTC-MSG [23] do not appear to apply to these critically ill patients. Other recommendations were developed to differentiate between colonization and infection [24]; nevertheless, it seemed reasonable to assume that a critically ill, mechanically ventilated patient with deteriorating respiratory function, radiological pulmonary infiltrate and positive for galactomannan antigen in an invasive respiratory sample warranted treatment without delay or needing to wait for culture results. Concerned about the high incidence and associated mortality observed at the beginning of the pandemic, the local clinical team considered the possibility of treating early with antifungal therapy, despite the fact that it was an off-label use, by including in the clinical decision, discontinuation of treatment if the mycological results of a good quality respiratory sample were negative.

Our findings may provide an opportunity to design a randomized study of higher methodological quality that would clarify the debate on whether to initiate treatment early based on such high suspicion and documented incidence [25].

The clinical argument for initiating treatment in these critically ill patients is that it is well known that a delay in targeted treatment of serious infectious diseases generally [26], and especially in aspergillosis, has been shown to increase mortality [27]. At the same time, the risk/benefit balance of the safety profile offered by current antifungal treatments [28] tends to favor treatment. Renal function was monitored in all patients administered liposomal amphotericin B treatment at a dose of 3 mg/kg/day, and no treatment-related adverse effects were observed. Treatment duration was a few days, the time it took to receive the mycological results.

The fact that we investigated aspergillus infection with the invasive bronchoscopy strategy only in intubated patients raises the possibility that we were probably underdiagnosing this respiratory coinfection in patients with SARS-CoV-2 pneumonia who did not undergo mechanical ventilation. The reason we did this was that the clinical status of those receiving high-flow oxygen therapy or noninvasive ventilation was such that we thought that the risk associated with bronchoscopy could lead to having to intubate patients or worsen their already impaired respiratory function.

ARDS may be an independent risk factor for pulmonary aspergillosis. As has been hypothesized for ARDS caused by influenza A [5], viral infection is thought to cause a state of lymphocyte dysfunction [29] that favors the possibility of fungal damage, in addition to the alveolar damage caused by viral aggression, and compartmentalization of the inflammatory response in the lung, which may favor the pathogenic capacity of the *Aspergillus* species. To these possible pathological mechanisms, we should add the effects of SARS-CoV-2 treatments. Despite the inconclusive results of immunomodulatory treatments [30], these continue to be administered in clinical practice; indeed, some authors have shown that they favor fungal infections [31]. The clear link between the administration of corticosteroids and the development of pulmonary aspergillosis is not disputed [32]; however, corticosteroid treatment has become widespread in critically ill patients with SARS-CoV-2, as the results have shown a reduction in mortality from 41.4% to 29.3% [33].

The debate as to whether colonization or infection is involved, as a means of deciding whether to treat early before the mycological results are available, will not be resolved by upper respiratory tract sampling as the clinical features are often nonspecific. For this reason, we recommend targeted lower respiratory tract sampling, in which isolation of *Aspergillus* species in essentially sterile specimens does not give rise to doubts about the possibility of infection in critically ill patients.

The main strength of the present study is that it is an observational study based on actual clinical practice, prompted by the clinicians’ need to combat the high incidence of aspergillosis coupled with high mortality in the early months of the pandemic. There is experience and studies on fungal infection prophylaxis in hematologic patients [34], but there is a notable paucity of publications on the subject in critically ill patients, as highlighted by the recent clinical trial with posaconazole conducted in 88 patients with respiratory distress syndrome due to influenza [35]. The lack of optimal results in reducing the incidence or improving mortality led us to decide to perform early antifungal therapy, but using a different antifungal treatment, in this case liposomal amphotericin B.

Using this novel strategy, we documented a decrease in the incidence of pulmonary aspergillosis in patients with severe SARS-CoV-2 pneumonia, and also observed an improvement in mortality that could be justified by the early anticipatory treatment. The invasive diagnostic strategy using fibrobronchoscopy ensured safe and early withdrawal of antifungal treatment when a value of <0.5 was obtained for the galactomannan assay due to its high negative predictive value. The different virus variants during the various waves of the pandemic were not analyzed, which is a factor that may have influenced the susceptibility to develop pulmonary aspergillosis, but this was not studied. It is well documented that vaccination considerably reduced the number of admissions to the ICU, and that the dominant profile in these admissions was that of the immunocompromised patient [36]. However, as is evident from the results, the percentage of vaccinated patients admitted to the ICU was a minority. The hypothesis that, as time passed during the pandemic, viruses tended to decrease in virulence, which might play a role in conferring lower susceptibility to pulmonary aspergillosis among patients, has not actually been demonstrated.

The main limitation of this study is that it describes the clinical experience of a single center, and it would be useful to check whether our results were reproduced in other ICUs. We compared two study periods at a time as turbulent as the SARS-CoV-2 pandemic when there were notable changes in treatment strategies, and the data described in the study results should be interpreted in this sense. In the second study period, infection was caused by other variants, and the profile of the critically ill patient changed to include vaccinated patients, who were a minority in our population. We do not know what impact this would have on the incidence of aspergillosis. The new clinical profile of the vaccinated patient admitted to the ICU is frequently one of immunosuppression [37], which is itself a risk factor for aspergillosis. However, we report some novel aspects that are worth mentioning. Fibrobronchoscopy was performed in most patients included in the study; these patients were mechanically ventilated in prone position with severe deterioration of respiratory function [36]. The procedure was safe and cost-effective. The incidence of pulmonary aspergillosis using an active diagnostic search and invasive diagnostic bronchoscopy was high. A decrease in the incidence of pulmonary aspergillosis was observed with the early antifungal therapy strategy, with no adverse effects due to the short duration of treatment in patients who did not have aspergillosis. Finally, despite the improvements in the antifungal arsenal and diagnostic tools, pulmonary aspergillosis remains a serious coinfection with high preventable mortality in SARS-CoV-2 patients. Assuming that these patients are an at-risk group may help to detect this infection early and treat it promptly to improve the clinical outcome of aspergillosis in critically ill patients.

## Figures and Tables

**Figure 1 jof-09-00288-f001:**
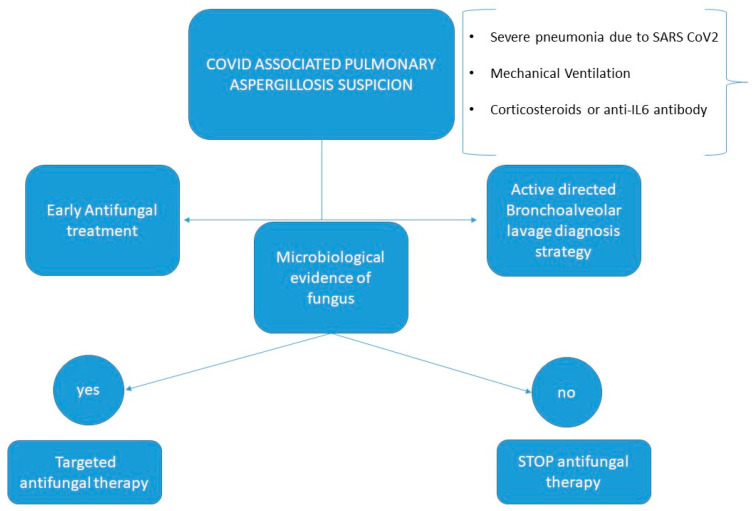
Block diagram showing the algorithm to guide clinical decision during the second period of the study.

**Figure 2 jof-09-00288-f002:**
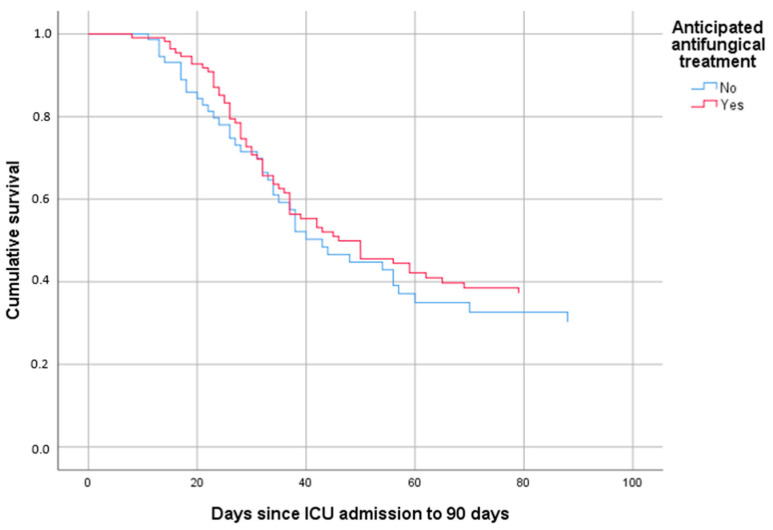
Overall mortality at 90 days from ICU admission in patients with SARS-CoV-2 infection with and without early antifungal treatment at ICU admission.

**Figure 3 jof-09-00288-f003:**
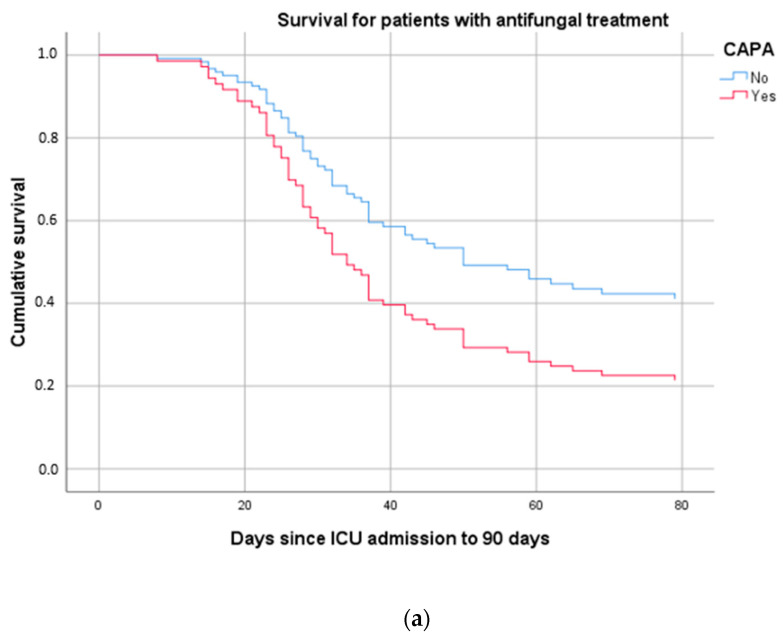
(**a**) Mortality at 90 days from ICU admission in patients with SARS-CoV-2 infection and early antifungal treatment at ICU admission. (**b**) 90-day mortality from ICU admission in patients with SARS-CoV-2 infection without early antifungal treatment at ICU admission.

**Table 1 jof-09-00288-t001:** Baseline characteristics of the study population on admission to the ICU.

Variable	*n*	Overall (*n* = 160)	No Antifungal Treatment (*n* = 58)	Anticipatory Antifungal Treatment (*n* = 102)	*p*
Demographic variables					
Age (years)	160	65 [57–71]	64 [54–71]	65 [58–71]	0.687
Female sex	160	45 (28%)	13 (22%)	32 (31%)	0.226
BMI (kg/m^2^)	94	30.4 [26.6–33.2]	27.8 [26.9–33]	31 [28.3–33.4]	0.128
Coexisting conditions					
Heart disease	160	26 (16%)	10 (17%)	16 (16%)	0.798
Metabolic disorder	160	16 (10%)	3 (5%)	13 (13%)	0.125
Hypertension	160	99 (62%)	30 (52%)	69 (68%)	0.046
Diabetes	160	58 (36%)	20 (35%)	38 (38%)	0.726
Obesity (BMI > 30)	160	51 (32%)	13 (22%)	38 (38%)	0.048
Liver disease	160	9 (6%)	3 (5%)	6 (6%)	0.851
Chronic kidney disease	160	13 (8%)	1 (2%)	12 (12%)	0.027
COPD	160	6 (4%)	1 (2%)	5 (5%)	0.309
Asthma	160	7 (4%)	2 (3%)	5 (5%)	0.666
Active malignancy	160	10 (6%)	6 (10%)	4 (4%)	0.10
Chronic neurological disease	160	11 (7%)	2 (3%)	9 (9%)	0.196
Prior transplantation	160	7 (4%)	1 (2%)	6 (6%)	0.216
Immunosuppression	160	2 (1%)	0 (0%)	2 (2%)	0.283
SARS-CoV-2 Vaccination	160	7 (4.37%)	0 (0%)	7 (6.86%)	0.049

Data are presented as absolute counts (%) or as medians [25th–75th percentiles]. Abbreviations: BMI: body mass index; ICU: intensive care unit; COPD: chronic obstructive pulmonary disease.

**Table 2 jof-09-00288-t002:** Laboratory tests of patients on admission to the ICU and on the third day after admission.

Variable	*n*	Overall (*n* = 160)	No Antifungal Treatment (*n* = 58)	Early Antifungal Treatment (*n* = 102)	*p*
Laboratory values					
*Admission to ICU*					
LDH (U/l)	142	533 [387–703]	510 [389–633]	563 [386–718]	0.271
Creatinine (mg/dl)	158	0.9 [0.7–1.2]	0.9 [0.7–1.2]	0.8 [0.7–1.1]	0.793
CRP (mg/L)	157	98 [28–194]	123 [34–231]	81 [19–173]	0.049
Procalcitonin (ng/mL)	156	0.14 [0.07–0.43]	0.21 [0.08–0.68]	0.12 [0.06–0.29]	0.022
Total bilirubin (mg/dl)	93	0.6 [0.4–0.9]	0.6 [0.4–1.0]	0.7 [0.4–0.9]	0.687
Ferritin (ng/mL)	130	1354 [627–2322]	1642 [681–2542]	1334 [605–2324]	0.520
D-dimer (mg/l)	149	1.7 [0.9–7]	1.2 [0.7–3.3]	2.5 [1.2–12.4]	<0.0001
IL-6 (pg/mL)	18	36 [19–294]	72 [23–944]	23 [17–208]	0.328
*Blood counts*					
-Leukocytes [G/I]	157	11.2 [7.6–15]	10.2 [7.4–14.5]	11.9 [7.9–15.7]	0.294
-Neutrophils [G/I]	157	10.2 [6.9–14]	9.2 [6.4–12.9]	10.6 [7.1–14.6]	0.174
-Lymphocytes [G/I]	157	0.6 [0.5–0.9]	0.7 [0.5–0.9]	0.6 [0.4–1.0]	0.928
-Thrombocytes [G/I]	157	241 [180–313]	240 [172–310]	243 [182–320]	0.785
*Third day of ICU admission*					
LDH (U/l)	125	450 [334–614]	473 [318–610]	447 [339–615]	0.913
Creatinine (mg/dl)	157	0.9 [0.7–1.2]	0.9 [0.7–1.3]	0.9 [0.7–1.2]	0.776
CRP (mg/l)	153	82 [25–172]	158 [78–229]	42 [13–112]	<0.0001
Procalcitonin (ng/mL)	155	0.2 [0.1–0.5]	0.3 [0.1–1.3]	0.1 [0.1–0.4]	0.003
Bilirubin total (mg/dl)	103	0.7 [0.4–1.0]	0.7 [0.4–1.0]	0.6 [0.4–0.9]	0.149
Ferritin (ng/mL)	113	1341 [594–2617]	1573 [732–3815]	1209 [509–2282]	0.074
D-dimer (mg/l)	145	2.1 [1.1–6.6]	1.7 [0.9–5.4]	2.6 [1.3–9.7]	0.026
IL-6 (pg/mL)	13	66 [36–1076]	192 [55–1448]	39 [11–184]	0.090
*Blood counts*					
-Leukocytes [G/I]	158	11.5 [8.5–15.5]	10.7 [8.2–14.7]	12 [8.7–16.2]	0.209
-Neutrophils [G/I]	158	9.8 [7.2–13.7]	9.4 [6.7–12.6]	10.2 [7.5–14]	0.235
-Lymphocytes [G/I]	154	0.7 [0.4–1.0]	0.7 [0.5–0.9]	0.7 [0.4–1.0]	0.681
-Thrombocytes [G/I]	158	243 [174–316]	258 [181–337]	239 [169–303]	0.255

Data are reported as absolute counts (%) or as medians [25th–75th percentile]. Data are reported as absolute counts (%) or as medians [25th–75th percentiles]. Abbreviations: IL-6: interleukin 6; CRP: C-reactive protein.

**Table 3 jof-09-00288-t003:** Incidence of pulmonary aspergillosis and outcome in the study population.

Variable	*n*	Overall (*n* = 160)	No Antifungal Treatment (*n* = 58)	Anticipatory Antifungal Treatment (*n* = 102)	*p*-Value
Outcomes					
CAPA (probable)	160	30 (19%)	19 (32.75%)	11 (10.78%)	0.001
Length of ICU stay	160	19 [11–29]	17 [11–30]	19 [11–29]	0.790
90-day mortality	160	103 (64%)	41 (71%)	62 (61%)	0.282
CAPA (probable) mortality 90-day mortality in CAPA	30	26 (87%)	19 (100%)	7 (63.63%)	0.015

Data are reported as absolute counts (%) or as medians [25th–75th percentiles]. *p* values are from the Mann–Whitney U test, Pearson’s χ2 test, Fisher’s exact tests or the Log-rank test, as appropriate. Abbreviations: CAPA: coronavirus disease-associated pulmonary aspergillosis.

**Table 4 jof-09-00288-t004:** Comparative data according to CAPA diagnosis and early antifungal therapy.

		IPA = YES	IPA = NO	*p* Value
D-dimer at ICU admission (mg/L)	Anticipatory treatment	1464 [1091–14689.5]	3438 [1260–12,872.5]	0.612
	No anticipatory treatment	1272 [969.75–3962.25]	1254 [669–2791]	0.360
CRP at ICU admission (mg/L)	Anticipatory treatment	31.1 [10.8–186.9]	85.7 [22.2–171.625]	0.538
	No anticipatory treatment	136.75 [61.13–200.23]	99.95 [30.68–234.68]	0.958
Procalcitonin at ICU admission (ng/mL)	Anticipatory treatment	0.09 [0.06–0.16]	0.12 [0.06–0.2975]	0.858
	No anticipatory treatment	0.275 [0.0875–1.5825]	0.17 [0.08–0.62]	0.381
APACHE II at ICU admission	Anticipatory treatment	10 [8–12]	10 [9–12]	0.777
	No anticipatory treatment	10 [8–12]	9 [8–11]	0.282
SOFA at ICU admission >6	Anticipatory treatment	0 (0%)	17 (60.71%)	0.687
	No anticipatory treatment	5 (45.46%)	5 (31.25%)	0.179
D-dimer at the beginning of mechanical ventilation (mg/L)	Anticipatory treatment	7422 [1019.5–17,691.25]	3784 [1583.5–11,447]	0.774
	No anticipatory treatment	2007.5 [1176.75–5665.75]	1167.5 [775.25–5384.5]	0.152
CRP at the beginning of mechanical ventilation (mg/L)	Anticipatory treatment	139.8 [14.25–188.7]	75.6 [19.1–145.6]	0.406
	No anticipatory treatment	145.5 [70.3–172]	116.25 [37.95–236.75]	0.791
Procalcitonin at the beginning of mechanical ventilation (ng/mL)	Anticipatory treatment	0.12 [0.0675–0.56]	0.115 [0.07–0.2925]	0.833
	No anticipatory treatment	0.35 [0.17–0.83]	0.14 [0.08–0.61]	0.183
SOFA at the beginning of mechanical ventilation >6	Anticipatory treatment	3 (27.27)	50 (54.95)	0.113
	No anticipatory treatment	15 (78.95)	19 (48.71)	0.046
Exitus	Anticipatory treatment	7 (63.64%)	55 (60.44%)	1.000
	No anticipatory treatment	19 (100%)	22 (56.41%)	0.000

## Data Availability

Not applicable.

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
