# Peer review of "Anticipatory Antifungal Treatment in Critically Ill Patients with SARS-CoV-2 Pneumonia"

_jof, 2023, doi:10.3390/jof9030288_

Round 1

Reviewer 1 Report

Estella et al. present a clinical pilot study of the effects of antifungal prophylaxis for the treatment of SARS COV2 pneumonia patients. The manuscript is well written and easily to read. It is technically sound and offers interesting new data which might improve the understanding of  CAPA development.

Nonetheless, I have two points which should be addressed by the authors prior to publication.

1) Vaccination against SARS COV2 started in late 2020, early 2021. That means that at least 18 months of the study duration overlap with the time of vaccination campagns. However, the authors did not include a statement how many patients in the study were vaccinated  prior to hospital admission.  This should be added, e.g. in table 1. They should also mention if vaccinated individuals got 1, 2 or 3 doses of the vaccine.

2) I think that liposomal amphotericin B is an uncommon choice for antifungal prophylaxis. Are there any kind of data of a prophylaxis with voriconazole or posaconazole which are more common for anti-Aspergillus prophylaxis (e.g. stem cell transplantation patients). How were the longterm effects of amphotericin B treatment?

Author Response

Dear reviewer. We very much appreciate your comments on the manuscript. Thank you for your comments that will certainly improve the quality of the manuscript. Below we respond point by point to your suggestions and make improvements to the main text of the manuscript.

  • According with your suggestion we have added in the table 1 the percentage of vaccinated patients in the second time of the study.

We observed a significant decrease in ICU admissions with the implementation of vaccination, however unvaccinated patients continued to be admitted in ICU. Most of ICU admission in this time were for unvaccinated patients.

In accordance with your recommendation we have added text regarding vaccination in the study limitations in the Discussion section:

In the second period studied, the infection is caused by other strains, and the profile of the critical patient has been modified, including vaccinated patients who were in the minority, due we do not know the impact that this will have on the incidence of aspergillosis documented in studies published during the first year of the pandemic.

  1. In reference to your commentary, the discussion contemplates the use of other antifungals in the critical patient and we justify the choice of clinical amphotericin B.

          “…the clinical trial conducted in 88 patients with respiratory distress

           syndrome due to influenza conducted with posaconazole (34). The lack

          of optimal results in reducing incidence or improving mortality led us to

          decide to perform anticipated antifungal therapy with another antifungal

           treatment such as liposomal amphotericin B.”

Regarding your question about how were the longterm effects of amphotericin B treatment?

In discussion section we comment: Renal function was monitored in all

liposomal amphotericin B treatments administered at doses of 3

mg/kg/day with no treatment-related adverse effects observed. It is true

that the duration of treatment was a few days waiting for mycological

results.”

English language and style has been reviewed. We provide a certificate from a professional English translation Company.   

Thank you for your recommendations that contribute to improving the quality of the manuscript.

Reviewer 2 Report

This study by Ángel Estella et al aims to investigate the incidence of COVID-19-associated pulmonary aspergillosis (CAPA) in critically ill patients and whether anticipated antifungal treatment reduces the incidence of CAPA in critically ill patients. However, the results are incomplete and the conclusion can’t be made based on the current study design. My major concerns are as follows:

1. This study included patients mainly from two periods (from 2020 to 2022). However, the virus spectrum differs largely over these periods. In addition, the pathology ability of these different strains also differs a lot. As shown in table 2, CRP, PCT, and D-D were significantly higher in no antifungal treatment patients who were included mainly in the first period, when the virus (mostly Alpha, Beta, and Delta) were much more harmful and caused more death. Therefore, it is hard to tell whether the low incidence of CAPA was due to the intern difference of the variant itself or the intervention of antifungal treatment. Therefore, the incidence of CAPA should also be compared in the same period between patients receiving antifungal treatment and not receiving it.

2. Any use of corticoids or immune suppression agents before admitted to the ICU should be clarified, as these treatments will interfere with the incidence of CAPA.

Author Response

Dear reviewer. We very much appreciate your comments on the manuscript. Thank you for your suggestions that will certainly improve the quality of the manuscript. Below we respond point by point to your suggestions and make improvements to the main text of the manuscript.

  1. No doubt its appreciation is a limitation of the study, in fact we contemplate it in the discussion of the manuscript in the discussion section.

Limitation. “In the second period studied, the infection is caused by other strains, and the profile of the critical patient has been modified, including vaccinated patients, who were in the minority. We do not know the impact that this will have on the incidence of aspergillosis                         documented in studies published during the first year of the pandemic.”

On the other hand, in such complicated times in the ICU and microbiology laboratories, the determination of the virus strains was not done routinely.

As this is a real clinical practice study, we regret not having carried out a design like the one proposed by you with two treatment arms within the same period that would undoubtedly have made the groups compared more homogeneous.

We explain in the manuscript the motivation that led us to initiate an early treatment strategy as clinicians in real clinical practice.

Without a doubt, your suggestion would have meant an improvement in the quality of the methodological design of the study. Thank you for your appreciation.

  1. Any use of corticoids or immune suppression agents before admitted to the ICU should be clarified, as these treatments will interfere with the incidence of CAPA.

According with your suggestion results section includes this data: “With respect to the concomitant treatments received, we observed significant differences in the two periods analyzed regarding the use of corticoids, 78% in patients not treated versus 94% in the group of patients with anticipated antifungal therapy. 3% of patients without antifungal therapy received treatment with tocilizumab versus 27% in the group of patients with anticipated antifungal therapy.”

English language and style has been reviewed. We provide a certificate from a professional English translation Company. 

Reviewer 3 Report

Review:

Background

The main objective of this study was to evaluate whether critically ill patients, when making early use of Amphotericin B, promote a lower incidence of pulmonary aspergillosis associated with COVID-19 (CAPA). The strategy was to use an invasive method to collect samples from the exact location of the pulmonary involvement identified by X-ray, associated with a microbiological detection method of the fungus to define the fungal infection.

Minor concerns

The authors present interesting and highly impactful results for critically ill patients affected by COVID-19. Three points deserve to be discussed in more depth:

1. Regarding the periods analyzed in the article, the first from March to November 2020 and the second from November 2020 to June 2022. The major concern that was not well discussed in the article is that in these two periods it is known that the circulating variants of SARSCOV2 were different, in the first waves of COVID19, the circulating variant had more severe disease than the variants that appeared in the other subsequent waves. Thus, the authors should include a more detailed analysis of the data due to the waves of COVID19 and variants circulating in the period, since the more recent cases, being less severe, may interfere with the result obtained by the authors of a decrease in the incidence of CAPA and associated mortality after use of antifungal, in relation to the second period analyzed.

2. The second important point is that the article needs to undergo an extensive review in English before publication. Many grammatical errors were found throughout the text.

3. Regarding the use of the antifungal, the authors must include the conditions of use, such as dose, type of amphotericin B and extension of treatment (for patients with CAPA), in the materials and methods, and not just in the discussion section of the work.

Author Response

Dear reviewer. We very much appreciate your comments on the manuscript. Thank you for your comments that will certainly improve the quality of the manuscript. Below we respond point by point to your suggestions and make improvements to the main text of the manuscript.

  1. Thank you for your comments, I agree with your assessment. Unfortunately, the determination of virus variants in patients admitted to the ICU was not a daily practice in the ICU. The pressure on microbiology laboratories with a high demand for work in the worst months of the pandemic prevented that more exhaustive analysis.

Your comment has been contemplated in the discussion of the article and is one of the main limitations of our study that must be taken into account in the interpretation of our results. We have also included in the results the low percentage of patients who were admitted to the ICU while vaccinated. Finally, based on your commentary, we include in the discussion the hypothesis that as time passes during the pandemic, viruses tend to decrease their virulence and this fact may contribute to confer a lower susceptibility of patients to pulmonary aspergillosis. Ç

We have added this comment in the discussion which is certainly of great interest.

“ We did not analyze the different variants of virus during the waves of the pandemic, this is a factor that has been able to influence the susceptibility to present pulmonary aspergillosis but it is not studied, it is documented that vaccination considerably decreased the number of admissions in ICU and that the profile of immunocompromised patients (35) predominated in these admissions,  however, as we can see in the results, the percentage of vaccinated patients admitted to the ICU was a minority.”

the hypothesis that as time passes during the pandemic, viruses tend to decrease their virulence and this fact may contribute to confer a lower susceptibility of patients to pulmonary aspergillosis”

  1. According with your recommendation an extensive review in english has been reviewed.

  1. According with your suggestion, the use, dose and type of amphotericin B has been included in the material and methods section.

English language and style has been reviewed. We provide a certificate from a professional English translation Company. 

Thank you for your recommendations that contribute to improving the quality of the manuscript.

Author Response

Dear reviewer. We very much appreciate your comments on the manuscript. Thank you for your comments that will certainly improve the quality of the manuscript.

We have made the changes suggested in your review and certainly consider that the manuscript has undergone substantial improvements.

  • Publishes guidelines cited in introduction section has been added the text.

[2] Patterson TF, Thompson GR 3rd, Denning DW, Fishman JA, Hadley S, Herbrecht R, Kontoyiannis DP, Marr KA, Morrison VA, Nguyen MH, Segal BH, Steinbach WJ, Stevens DA, Walsh TJ, Wingard JR, Young JA, Bennett JE. Practice Guidelines for the Diagnosis and Management of Aspergillosis: 2016 Update by the Infectious Diseases Society of America. Clin Infect Dis. 2016 Aug 15;63(4):e1-e60. doi: 10.1093/cid/ciw326. Epub 2016 Jun 29. PMID: 27365388; PMCID: PMC4967602.

  • We have not referred to the name of the hospital on material and methods for maintaining a blind revisión of the manuscript.
  • Grammatical errors have been corrected, thank you for your review and note them.
  • Information about kit used to measure galactomannan has been added in material and methods section according with you suggestion:

“Bronchoalveolar lavage (BAL) specimens were cultured in sabouraud agar with chloramphenicol (Becton Dickinson) and Platelia Aspergillus galactomannan assay (Bio-Rad Laboratories) were assessed.”

  • Dose of antifungal therapy has been mentioned in the text y material and methods section according with your suggestion.
  • According with your suggestions we have added in result sections treatment related adverse effects in renal function.

English language and style has been reviewed. We provide a certificate from a professional English translation Company. 

Thank you for your recommendations that contribute to improving the quality of the manuscript.

Round 2

Reviewer 2 Report

No comments